# Phenotypic Characterization and Draft Genome Sequence Analyses of Two Novel Endospore-Forming *Sporosarcina* spp. Isolated from Canada Goose (*Branta canadensis*) Feces

**DOI:** 10.3390/microorganisms12010070

**Published:** 2023-12-29

**Authors:** Jitendra Keshri, Kristina M. Smith, Molly K. Svendsen, Haley R. Keillor, Madeline L. Moss, Haley J. Jordan, Abigail M. Larkin, Johnna K. Garrish, John Eric Line, Patrick N. Ball, Brian B. Oakley, Bruce S. Seal

**Affiliations:** 1College of Veterinary Medicine, Western University of Health Sciences, Pomona, CA 91766, USA; jitendrakeshri2012@gmail.com; 2Biology Program, Oregon State University Cascades, Bend, OR 97702, USA; kristina.smith@osucascades.edu (K.M.S.); themollysvendsen@gmail.com (M.K.S.); haleyrosemmk@gmail.com (H.R.K.); mossm@oregonstate.edu (M.L.M.); hjanejordan@gmail.com (H.J.J.); abilarkin@gmail.com (A.M.L.); pat.ball@osucascades.edu (P.N.B.); 3Poultry Microbiological Safety & Processing Research Unit, U.S. National Poultry Research Center, Athens, GA 30605, USA; johnna.garrish@usda.gov (J.K.G.); ericline9@gmail.com (J.E.L.)

**Keywords:** microbiome, spore-forming bacteria, urease, probiotic, Firmicutes, *Planococcaceae*

## Abstract

In an attempt to isolate new probiotic bacteria, two Gram-variable, spore-forming, rod-shaped aerobic bacteria designated as strain A4 and A15 were isolated from the feces of Canada geese (*Branta canadensis*). Strain A4 was able to grow in high salt levels and exhibited lipase activity, while A15 did not propagate under these conditions. Both were positive for starch hydrolysis, and they inhibited the growth of *Staphylococcus aureus*. The strains of the 16S rRNA sequence shared only 94% similarity to previously identified *Sporosarcina* spp. The ANI (78.08%) and AAI (82.35%) between the two strains were less than the species threshold. Searches for the most similar genomes using the Mash/Minhash algorithm showed the nearest genome to strain A4 and A15 as *Sporosarcina* sp. P13 (distance of 21%) and *S. newyorkensis* (distance of 17%), respectively. *Sporosarcina* spp. strains A4 and A15 contain urease genes, and a fibronectin-binding protein gene indicates that these bacteria may bind to eukaryotic cells in host gastrointestinal tracts. Phenotypic and phylogenetic data, along with low dDDH, ANI, and AAI values for strains A4 and A15, indicate these bacteria are two novel isolates of the *Sporosarcina* genus: *Sporosarcina* sp. A4 sp. nov., type strain as *Sporosarcina cascadiensis* and *Sporosarcina* sp. A15 sp. nov., type strain *Sporosarcina obsidiansis*.

## 1. Introduction

The genus *Sporosarcina* was first proposed by Kluyver and van Niel (1936) [1] in the family Bacillaceae. Currently, this genus includes at least 17 species with valid published names (www.bacterio.net/sporosarcina.html (accessed on 6 April 2020), isolated from diverse environments such as industrial clean room floors and human blood samples [2], soils [3,4,5,6], sea sediment [7], surface water and sediments of a brackish lake [8], seawater [9], clinical specimens and cow’s milk [10], cyanobacterial pond mat [11], and equipment for soy sauce production [12]. Interestingly, during a ground-based simulated microgravity (SMG), a mouse model, *Sporosarcina* spp., was used as a member of the microbiome as a biomarker during SMG [13], and members of the genus were isolated from the environmental surfaces of the International Space Station [14].

Industrial uses for members of the genus *Sporosarcina* include microbially induced calcium carbonate precipitation during construction and geotechnical applications [15], ureolytic biomineralization [16], and as potential microorganisms for soil improvement [17]. The predominant use for these bacteria is for microbially induced calcite precipitation by ureolysis during constructional engineering and material applications [18]. Phenotypically, the species belonging to this genus are either Gram-positive or Gram-variable and morphologically rod or spherical in shape. The *Sporosarcina* spp. members form endospores and are heterotrophic [19]. The major isoprenoid quinone and cellular fatty acid are MK-7 and anteiso-C15:0, respectively [4]. The G+C content of the genomic DNA varies from 39 to 46.5 mol% [7].

Although most uses for *Sporosarcina* spp. are industrial, three members of the genus, *S. pasteurii*, *S. globispora*, and *S. psychrophila*, reportedly possess properties that make them potential poultry probiotics [20]. Many spore-forming, non-toxin-producing bacteria reportedly promote anti-inflammatory immune responses and promote the healthy development of the monogastric animal gut. Free-ranging as well as domestic bird species harbor diverse communities of microorganisms in their GI tract that play crucial roles in providing the host with nutrition and protection from pathogens similar to other animals such as the mouse [13]. Consequently, our hypothesis was that potentially beneficial bacterial cultures could be identified from free-ranging geese’s gastrointestinal material using simple benchtop procedures to select spore-forming bacteria. Based on previous results [20], we hypothesized that isolating chloroform-resistant bacterial strains from Canada goose (*Branta canadensis*) feces would be selected for spore-forming bacteria of avian species.

## 2. Materials and Methods

### 2.1. Isolation and Media Culture of Bacteria

Fresh feces from free-ranging geese were aseptically collected from resident Canada goose (*Branta canadensis*) populations in Bend, OR, USA (44.0582° N, 121.3153° W), utilizing sterile plastic bags during autumn of 2018. Fecal material was stored in an ultra-cold freezer (−80 °C) until processed. Goose fecal material was thawed and suspended in phosphate-buffered saline (PBS) using organic solvent-resistant polypropylene 15 or 50 mL conical centrifuge tubes followed by vortex mixing for five minutes [21]. Subsequently, low-speed centrifugation was conducted at 1000× *g* for five minutes to eliminate solids. Following centrifugation, chloroform was added to a concentration of approximately 3 percent, e.g., 0.3 mL per 10 mL of fecal suspension or 1.5 mL chloroform per 50 mL fecal suspension and placed on a laboratory shaker for 30 min to eliminate vegetative bacterial cells and select for bacterial spores [22,23]. Aliquots of spore suspensions (150 µL) were cultured aerobically for two days utilizing brucella agar with blood and vitamin K-hemin (BBHK) [24]. This procedure resulted in fifteen axenic aerobic bacterial isolates, of which two (designated A4 and A15) were selected for further analyses. Subsequently, the isolates were passaged on brain heart infusion (BHI) agar media [25]. The isolates were also plated on eosin-methylene blue (EMB) agar [26] and mannitol salt agar [27] media. Gram stains were completed utilizing basic bacteriological procedures [28], and the isolates were assayed by disc diffusion to streptomycin, erythromycin, chloramphenicol, penicillin, and tetracycline to assay for antibiotic resistances [29]. The isolates were also assayed for lipase activity using spirit blue agar plates [30] and for starch hydrolysis [31].

### 2.2. Growth Inhibition Assays for Antibacterial Analyses of Isolates

Growth inhibition assays were completed to determine antibacterial activity against *Staphylococcus aureus* [32]. Target bacteria and the putative spore-forming bacterial isolates were streaked from stocks on BHI agar (Becton Dickinson, Franklin Lakes, NJ, USA). Overnight cultures of both target bacteria *S. aureus* and the goose isolates A4 and A15 (2.5 mL) were propagated in liquid BHI media. Twenty-five µL of target bacteria (~10^6^ cells) were inoculated into 15 mL of sterile BHI agar that had been cooled to 55 °C. The inoculated agar was poured onto a sterile Petri dish and allowed to solidify under sterile conditions. The goose test bacterium was pelleted and suspended in 200 µL of BHI media, into which sterile filter discs were saturated with the test bacterium and then placed on the target bacterial agar plates. Inoculated plates with discs were incubated at 37 °C, and the formation of a zone of clearance (ZOC) was visually assessed after 24–36 h.

### 2.3. Phenotype Analyses of Isolates

Additional biochemical features were determined with the use of Phenotype Microarray plates (Biolog plates 9 and 10) [33,34,35], according to the manufacturer’s instructions. All the materials and reagents used were purchased from Biolog (Hayward, CA, USA). Average responses were calculated for PM9 and 10 MicroPlates, with results recorded after 72 h at 37 °C in an Omnilog incubator. For each well, the means of the two highest values were used for reported data. Values from wells that were deemed false positives according to growth observed in negative control plates (incubated without any inoculation) were not considered for analyses. To identify positive growth responses, we combined two analytical approaches. First, Tukey honest significant difference (HSD) tests were performed by comparing 72 h optical density (OD) values for each well against values of the negative control well. Second, 72 h OD values from each well were corrected by the values in the negative control and then normalized by the maximum mean value of the respective plate. Wells with significantly higher OD (*p* < 0.05) according to Tukey’s HSD tests and normalized values > 0.5 were considered as a positive (+) phenotypic response for the particular carbon source, and wells with normalized values > 0.7 were considered strong positives (++) for utilization of a carbon source [34,35].

### 2.4. Bacterial DNA Purification and Draft Whole Genome Sequencing

DNA was extracted from bacterial colonies with the QuickExtract™ Bacterial DNA Extraction Kit (Waltham, MA, USA) or the mBio UltraClean^®^ Microbial DNA Isolation Kit. Genomic DNA (Carlsbad, CA, USA) was used to amplify 16S rRNA genes using the primers 27F (5’-AGAGTTTGATCMTGGCTCAG-3’) and 1492R (5’-GGTTACCTTGTTACGACTT-3’) [36] with the SapphireAmp FAST PCR master mix (Clontech/Takara, Mountain View, CA, USA) for PCR. Two Gram-variable bacilli designated A4 and A15 had the most unique 16S rRNA sequences that were <94% similar to previously identified *Sporosarcina* spp. Genomic DNA was sent to the Bioinformatics Center at the Oregon State University, Center for Genome Research and Biocomputing (https://cgrb.oregonstate.edu/ (accessed on 18 April 2019), Corvallis campus, for whole genome sequencing using Illumina’s MiSeq platform 150 bp paired-end sequencing chemistry. Raw fastq files were used for de novo genome assembly with SPAdes v 3.10.1 [37]. Assembled scaffolds of strains A4 and A15 genome sequences were submitted to NCBI for BLAST analyses using Microbial genome Blast analysis tools [38]. Genome annotation was completed using the IMG/MER Pipeline [39]. Conserved domain analyses [40] and clusters of orthologous genes (COG) profiles [41] were also completed for both the A4 and A15 genomes.

### 2.5. Genome Sequence Analyses and Phylogenetic Analyses of Isolates’ Genomes

A search for similar genomes using the Mash/Minhash algorithm [42] showed the nearest genome to strain A4 and A15 as *Sporosarcina* sp. P13 (distance 21%) and *S. newyorkensis* (distance 17%), respectively. The Phyloflash pipeline [43] was also used for identification of genomes based on reconstructed 16S rRNA sequences from both the A4 and A15 genomes, revealing closest identity to an unknown species of the genus *Sporosarcina*. All pairwise comparisons among the set of genomes were conducted using Genome BLAST Distance Phylogeny approach (GBDP), while intergenomic distances and digital DDH values were calculated using the recommended settings of the GGDC 2.1 [44]. Average nucleotide identity (ANI) [45] and average amino acid identity (AAI) [46] were determined for 22 and 18 genomes, respectively, of the *Sporosarcina* spp. FastME 2.0 was used to infer phylogenies of A4 and A15 among the *Sporosarcina* spp., using a distance approach to produce a minimum evolution tree [47].

## 3. Results and Discussion

### 3.1. Phenotypic Characterization of Strains A4 and A15

Gram stains were completed, demonstrating that both isolates were Gram-variable in their staining (Figure 1). Although most *Sporosarcina* spp. are considered Gram-positive [19], there are reports of two other members of the genus, *S. luteola* [12] and *S. aquimarina* [48], that are reportedly Gram-variable. So, it is not unusual that isolates A4 and A15 stain are Gram-variable, although A4 and A15 were isolated from avian species rather than plant material [12,48]. Additionally, the isolates were exposed to streptomycin, erythromycin, chloramphenicol, penicillin, and tetracycline with a resultant antibiotic inhibition of the growth of both bacteria (Appendix A).

The isolates were both catalase and oxidase positive, indicating that, like most aerobic and facultatively anaerobic bacteria, these bacteria neutralize the bactericidal effects of hydrogen peroxide [49]. Oxidase + (Ox+) detects that the bacterium has cytochrome c oxidase as a terminal enzyme in the respiratory chain [50]. The different carbon source utilization characteristics of strains A4 and A15 were assessed by plating on eosin-methylene blue (EMB) agar and mannitol salt agar (MSA) media. A4 was able to replicate on EMB and MSA, while A15 did not propagate on either media. Although bacterium A4 was able to be propagated on both EMB and MSA media, A15 would not grow on these media. Consequently, the A4 isolate can use lactose and sucrose as a carbon source present in the EMB agar media as well as mannitol present in the MSA media. Additionally, A4 can tolerate growth in the 7.5% concentration of sodium chloride, considered high salt, which is in MSA media (Appendix A). Both A4 and A15 replicated with starch agar, resulting in a positive reaction indicating probable amylase activity [31]. Interestingly, A4 was positive on spirit blue plates, indicating lipase activity, whereas A15 was not positive for lipase (Appendix A). Consequently, A4 is potentially capable of producing the lipase enzyme, while A15 most likely does not express lipase [30]. Although *Bacillus* spp. and *Geobacillus* spp. are important sources of lipases, it was recently reported that an isolate closely related to *S. ginsengisoli* expressed lipase activity and the ability to digest starch [51]. Most metabolic investigations involve the use of *Sporosarcina* spp. for microbially induced calcium carbonate precipitation [15], and transcriptomics was utilized to determine the utilization of nitrogen sources for *S. pasteurii* that does not reportedly utilize glucose as a carbon source [52]. Although the utilization of glucose as a carbon source was detected phenotypically and has been reported by another investigative group along with supplementing media with maltose, lactose, fructose, and sucrose for increased bacterial growth [53].

PM9 and PM10 were used for testing ionic/osmotic and pH effects, respectively (Appendix A). The average response for each well was calculated as the mean of the two highest values from at least three replicates per plate. The means of each well were normalized by the maximum mean value of the plate, values higher than 0.5 were considered positive (+), and values above 0.7 were considered strong positive (++) for PM9 and 10 (Appendix A). Strain A4 and A15 could tolerate 10% and 4% of NaCl, respectively. However, both of them showed optimum growth at 3% NaCl. A4 and A15 could tolerate potassium chloride up to 4% and 6%, respectively, while both isolates could tolerate 5% sodium sulfate. A4 did not grow in the presence of urea, ethylene glycol, sodium lactate, or sodium nitrite, but A15 grew with 1% of sodium lactate, 10% of ethylene glycol, 2% of urea, or 60 mM of sodium nitrite. Strain A4 and A15 could tolerate 200 mM and 100 mM sodium phosphate (pH 7), respectively, and 20 mM and 100 mM of ammonium sulfate (pH 8), respectively. Strain A4 grew in the presence of 12 other osmolytes with 6% NaCl, and A15 could grow with 2 osmolytes and 6% NaCl (Appendix A). Strain A4 growth was positive in a pH range of 8–10, while A15 grew in the pH range of 6–10. Both strains utilized 26 different amino acids, 7 amino acid-derived organic compounds, amine oxide, and urea, under the stress of pH 9.5 (Appendix A). Strain A4 could grow with three glycoside stressors, while strain A15 grew with eight different glycosides. A4 showed growth with stressors X-SO4, while A15 showed growth with both X-SO4 and X-PO4. Interestingly, *S.* sp. B5 reportedly increased in cell numbers and biomass in seawater at −5 °C [54].

### 3.2. Growth Inhibition Assays of Target Bacteria for Strains A4 and A15

Growth inhibition assays were completed to determine if either A4 or A15 could inhibit the replication of *Staphylococcus aureus* (Figure 2). Both isolates produced a zone of clearance (ZOC) when using *S. aureus* as a target organism, with isolate A4 producing a zone that was double the ZOC diameter produced by A15. The ability to inhibit the growth of competing bacteria is a characteristic of probiotic microorganisms, and mining microbial sources such as bacteria or yeast to identify new antimicrobial compounds is a common path to the discovery of new feed additives [55], as reported herein.

Currently, only one previous report for members of the *Sporosarcina* spp. to be utilized as a potential probiotic has been proposed for use in monogastric animals, specifically for poultry [20]. *S. pasteurii*, *S. globispora*, and *S. psychrophila* adhered to the intestinal lumen, tolerated a pH between 2 and 4, tolerated bile to 0.5%, and inhibited the growth of several bacteria, including *S. aureus* and *Salmonella typhi* [20]. The only other report of a member of the genus as a potential nutritional supplement is *S. aquimarina* MS4 that is used as an overwintering agent for the sea cucumber *Apostichopus japonicus* [56].

### 3.3. Genome Analyses of Strains A4 and A15

The contigs of the A4 strain had a total length of 3,718,715 nucleotides with 85.95% coding DNA bases and GC content of 44.02%. A total of 4077 genes were predicted, including 95.93% of protein-coding genes, 1.18% of regulatory and miscellaneous features, and 2.89% of RNA genes. The functions of 74.76% of protein-coding genes were predicted. The contigs of the A15 strain had a total length of 3,642,744 nucleotides with 86.07% of coding DNA bases with a GC content of 40.97%. A total of 3942 genes were predicted, including 95.92% of protein-coding genes, 1.32% of regulatory and miscellaneous features, and 2.77% of RNA genes. The functions of 74.2% of protein-coding genes were predicted, and conserved domain analyses [40] resulted in the identification of sporulation, urea oxidase, urease, and fibronectin-binding protein genes present in the genomes of the two isolates (Appendix A).

Genes identified included those encoding a related *Bacillus*/*Clostridium* GerA spore germination protein wherein spores recognize germinants through receptor proteins encoded by gerA family of operons, which includes gerA, gerB, and gerK as well as a related *Bacillus subtilis* stage V sporulation protein R gene (spoVR) involved in spore cortex formation. These are gene products necessary for sporulation and germination [57,58]. Also, a gene encoding the urease accessory protein UreG was identified and was one of the four accessory proteins of urease responsible for the decomposition of urea to form ammonia and carbon dioxide [59,60]. The fibronectin-binding protein A gene (FbpA) was also present in the genomes of both isolates. This family consists of the N-terminal region of the prokaryotic fibronectin-binding protein. While fibronectin has critical roles in eukaryotic cellular processes, such as adhesion, migration, and differentiation, it is also a substrate for the attachment of bacteria. Although often a virulence factor among pathogenic bacteria, probiotic attributes of *Lactobacillus fermentum* include efficient binding to mucin and fibronectin [61].

Comparisons of COG abundance between the genomes of A4 and A15 with closely related genomes identified differences in the abundance of some important COG categories and pathways (Appendix A). For example, genes involved in carbohydrate transport and metabolism, amino acid transport and metabolism, and transcription regulations were 1.5–2 times more abundant in A4 than other closely related *Sporosarcina* genomes. For genome A15, genes involved in carbohydrate transport and metabolism and amino acid transport and metabolism were approximately 1.5 times more abundant in its genome relative to other *Sporosarcina* spp. genomes. In the A4 genome, the MFS family permease (COG0477) was more abundant than reference genomes (Appendix A). A methyl-accepting chemotaxis protein was twice as abundant in A4 versus A15. A total of 169 COGs and 48 uncharacterized proteins were exclusive to the A4 strain compared to its closest relatives *S. newyorkensis* 2681 [GCA_000220335.1], *S. newyorkensis* DSM 23,543 [SRX2126154], *S. ureae* DSM 2281 [GCA_000425545.1], *S. ureae* P17a [GCA_002082015.1], *S. ureae* P37 [CP015349.1], and *S. ureae* S204 [GCA_002081995.1]. Fifty-five enzymes, including hydrolases, phosphoglycerate dehydrogenase, and urease, were also more abundant in the A4 genome (Appendix A). The abundance of COGs representing the MFS family permease and Methyl-accepting chemotaxis protein in A15 genomes showed similar profiles to the A4 strain compared to reference genomes (Appendix A). A total of 145 COGs excluding 45 uncharacterized proteins were exclusive to the A15 strain and its closest relatives *S. newyorkensis* 2681 [GCA_000220335.1], *S. newyorkensis* DSM 23,543 [SRX2126154], *S. ureae* DSM 2281 [GCA_000425545.1], *S. ureae* P17a, *S. ureae* P37 [GCA_002082015.1], and *S. ureae* S204 [GCA_002081995.1]. Compared to reference *Sporosarcina* spp. genomes, 59 enzymes, including hydrolases, pyruvate dehydrogenase (acetyl-transferring), and acetyl-CoA C-acetyltransferase were more abundant in the A15 genome (Appendix A).

The major facilitator superfamily (MFS) family permease COG (COG0477) for strain A4 consisted of 48 genes, including the DHA1 family (bicyclomycin/chloramphenicol resistance-like; multidrug resistance protein-like) MFS transporter, DHA2 family lincomycin resistance protein-like MFS transporter, EmrB/QacA subfamily drug resistance transporter, FSR family fosmidomycin resistance protein-like MFS transporter, multidrug resistance protein, OFA family oxalate/formate antiporter-like MFS transporter, putative MFS transporter, and the YNFM family putative membrane transporter. Strain A15 had 49 genes belonging to COG0477, consisting of all genes mentioned for A4 with an additional DHA2 family metal-tetracycline-proton antiporter-like MFS transporter that also includes the includes anhydromuropeptide permease AmpG (Appendix A). These systems are a family of secondary active membrane transporters involved in the exchange of solutes and energy metabolism, such as carbohydrate transport and metabolism, amino acid transport and metabolism, and inorganic ion transport and metabolism [62]. Importantly, genes in this group have been identified as molecular signatures for bacterial phylogenomics, including the *Sporosarcina* spp. [63].

Because *Sporosarcina* spp. has been proposed to be potential probiotics [20], and the isolates reported herein have growth inhibitory properties against *S. aureus* (Figure 2), the genomes were searched for genes that may encode potential antimicrobials (Appendix A). An ABC-type bacteriocin/lantibiotic exporter that contains an N-terminal double-glycine peptidase domain (COG2274) was located in the A4 genome but not in other genomes of closely related *Sporosarcina* spp. Seven ABC-type antimicrobial peptide transport systems and permease components (COG0577) were detected in the A4 and A15 genomes but not in the *Sporosarcina* spp. genomes searched that are closely related to A4 or A15. Also, a peptidoglycan hydrolase (amidase) enhancer domain (COG2385) was detected in the A4 and A15 genomes. Nine and seven metal-dependent amidase/aminoacylase/carboxypeptidase (COG1473) genes were found in the A4 and A15 genomes, respectively. One phage-related protein, the tail component (COG4723), was detected in A4 and A15 genomes, and a phage tail tape-measure protein (COG3941) was in A15 but not A4. However, no phage-related lytic proteins were detected in the genomes, although many of these genes encode what could be considered potential alternatives to standard antibiotics [55].

### 3.4. Phylogenetic Relationships among the Sporosarcina spp. with Isolates A4 and A15

The average nucleotide identity analyses of both draft genomes were compared with twenty-two genomes of *Sporosarcina* spp. and three other closely related genomes. Strain A4 and A15 have maximum ANI with *S. newyorkensis* (SRX2126154 and SRX059760) of 78.6% and 81%, respectively (Table 1). NCBI utilizes ANI to evaluate the taxonomic identity of prokaryotic genome assemblies in GenBank. The results of the ANI analysis revealed that both the A4 and A15 isolates could be potentially novel species based on accepted ANI species cutoffs below 95% [45]. All the other *Sporosarcina* spp. had ANI less than the type species *S. newyorkensis*, which is less than 78.6% and 81%, respectively. For example, both A4 and A15 are only 76.23% to 77.38% similar to *Sporosarcina urea* strains within the genus.

Digital DNA-DNA hybridization was performed with 18 closely related genomes [44], and no genomes were significantly similar to any other *Sporosarcina* spp. (Table 2). In particular, both A4 and A15 strains had the highest dDDH% with *S. newyorkensis* (GCF_000220335.1) of 23.5% and 27.4%, respectively. A dDDH% of 79–80% is considered the threshold for delineating subspecies [64,65], further supporting A4 and A15 as novel isolates of the genus *Sporosarcina*. For reference, dDDH values between *S. newyorkensis* 2681 (SRX059760) and *S. ureae* DSM 2281 (GCA_002081995.1) was 20.7%; between *S. pasteurii* strain BNCC 337,394 (GCA_004379295.1) and *S. ureae* DSM 2281 was 13.2%, *S. newyorkensis* 2681 and *S. pasteurii* strain BNCC 337,394 was 13.5%.

The obtained intergenomic distances using formula d5 [64] were used to infer a balanced minimum evolution tree [47], which demonstrated that strains A4 and A15 could not be assigned to any existing species cluster (Figure 3). Additionally, the average amino-acid identity (AAI) of the genomes was performed using protein fasta files with the AA profiler [46]. Isolate A4 was most similar to *S. newyorkensis* 2681 with AAI 85.2% and matched fractions of 65.2%, while A15 also had the nearest similarity to *S. newyorkensis* 2681 with AAI 87.1% and matched fractions of 66.3% (Figure 4). Annotated nucleotide and protein fasta files were analyzed for species identification on the EMBL webserver specI (http://vm-lux.embl.de/~mende/specI// (accessed on 19 September 2019) [46], based on 40 universal single-copy marker genes. This further supports the designation of strains A4 and A15 as novel isolates, with the closest relative *S. newyorkensis* 2681 having 87% and 88.8% average identity, respectively. Consistent with this and with the other analyses performed here, a whole-genome phylogeny as shown in Figure 3 and Figure 4 showed the A4 and A15 isolates were most similar to S. newyorkensis isolates, although bootstrap support in many cases was weak. The second group is composed of *Sporosarcina* spp., including *S. urea*, but also includes the strictly anaerobic and chemolithoautotrophic acetogenic bacterium *Clostridium scatologenes* along with a Gram-negative *Flavobacterium reichenbachii* strain.

Principal coordinate clustering based on COG profiles demonstrated that the A4 and A15 isolates clustered together with each other and separated from other members of the genus (Figure 5). The nearest-neighbor genomes with A4 and A15 were from *S. newyorkensis* isolates. The reference *S. newyorkensis* strains are Gram-positive; endospore-forming rods originally isolated from veterinary clinical specimens in New York State, USA, and from raw milk in Flanders, Belgium, that were considered as new species of the genus [10]. *S. ureae* is a cocci-shaped [66] bacterium with a reported genome of 3.4 Mb and 41.5% GC content that has been isolated from soil contaminated with chicken waste [67]. *S. globispora* is reportedly a “round” Gram-positive, spore-forming, psychrophilic bacterium with a 5.67 Mb genome [68] that was originally classified as an endospore containing *Bacillus* spp. [9]. Other phylogenetically related species include the urease-producing bacterium *S. koreensis* with a 44.5% GC and 4.3 Mb genome [69] described as a Gram-positive, aerobic, spore-forming rod isolated from soil [6]. *S. psychrophila*, also a Gram-positive, spore-forming psychrophilic bacterial strain that is widely distributed in both terrestrial and aquatic environments had a reported circular chromosome of 4,674,191 bp with a GC content of 40.3%, also encoding urea hydrolysis genes [70]. A newly identified group of 28 cocci *Sporosarcina* spp. isolates had an average genome size of 3.3 Mb with two clades, one with 85.9% AAI and clade 2 with only 81.5% AAI to the other cocci-shaped *Sporosarcina* spp. indicating potentially new bacterial species [71]. *S. pasteurii* (~3.3 Mb genome, 39.2% GC; NZ_CP038012.1).

Other *Sporosarcina* spp. members have been proposed to be potential probiotic bacteria for poultry during production [20]. This includes the reduction of phytate by *Sporosarcina* spp. [72], and useful phosphatase genes (e.g., COG2194, COG0494, COG0561) were detected in the genomes of the isolates reported herein.

## 4. Summary and Conclusions

The results reported herein support the hypothesis that the A4 and A15 isolates are unique among the *Sporosarcina* spp. *S. cascadensis* A4 (cascadiensis. L. masc. adj. cascadensis referring to the area from which the isolates were obtained in the Cascade mountain range). The cells are Gram-variable, rod-shaped, occurring alone or in pairs. Colonies grown on BBHK or BHI agar are morphologically rods. Cells are aerobic, catalase-positive, and positive for oxidase test reaction with tetramethyl-p-phenylenediamine. The optimum growth temperature is 37 °C, optimum NaCl concentration for growth is 3% with a 10% maximum. Alkaliphilic with a pH growth range of 8–10. Acetic acid, acetoacetic acid, N-acetyl-d-glucosaminitol, Sec-butylamine, butyric acid, α-keto-butyric acid, d-fucose, maltotriose, mannan, oxalic acid, pectin, and l-sorbose are utilized individually as sole carbon sources for growth. The isolate A4 was isolated from the feces of the Canada goose (*Branta canadensis*). The DNA G+C content of the type strain is 44.02%. The type strain of *Sporosarcina cascadiensis* sp. nov. (A4) is held by the U.S. Department of Agriculture Agricultural Research Service (ARS) Culture Collection under the number: NRRL Y-65557.

*Sporosarcina obsidiansis* A15 (obsidiansis L. masc. adj. obsidiansis) refers to the geographic area from where the isolates were recovered that is high in obsidian rock. Cells are Gram-stain-variable, rod-shaped, occurring singly or in pairs. Colonies grown on BBHK or BHI agar are morphologically rods, aerobic, catalase-positive, and oxidase test reaction with tetramethyl-p-phenylenediamine positive. The optimum growth temperature is 37 °C, the optimum NaCl concentration for growth is 3%, the maximum NaCl concentration of 4%, pH growth ranges from 6 to 10. Carbon sources: d-alanine, l-alanine, l-asparagine, α-hydroxy butyric acid, α-keto-butyric acid, dulcitol, d-fructose, d-fructose-6-phosphate, fumaric acid, d-galactonic acid γ-lactone, l-galactonic acid γ-lactone, d-galactose, d-galacturonic acid, α-d-glucose, d-gluconic acid, d-glucosaminic acid, d-glucuronic acid, glucuronamide, α-hydroxy glutaric acid γ-lactone, glycolic acid, m-inositol, l-lactic acid, d,l-malic acid, maltotriose, d-mannitol, d-mannose, maltose, d-melibiose, mucic acid, p-hydroxy phenyl acetic acid, 1,2-propanediol, d-psicose, d-sorbitol, m-tartaric acid, d-threonine, l-threonine, tricarballylic acid, tyramine are utilized individually as sole carbon sources for growth. The strain, A15, was isolated from the feces of the Canada goose (*Branta canadensis*). The DNA G+C content of the type strain is 40.97%. The type strain of *Sporosarcina obsidiansis* sp. nov. (A15) is held by the U.S. Department of Agriculture Agricultural Research Service (ARS) Culture Collection under the number: NRRL Y-65558. *Sporosarcina obsidiansis* A15 (obsidiansis L. masc. adj. obsidiansis referring to the geographic area from where the isolates were recovered that is high in obsidian rock).

## Figures and Tables

**Figure 1 microorganisms-12-00070-f001:**
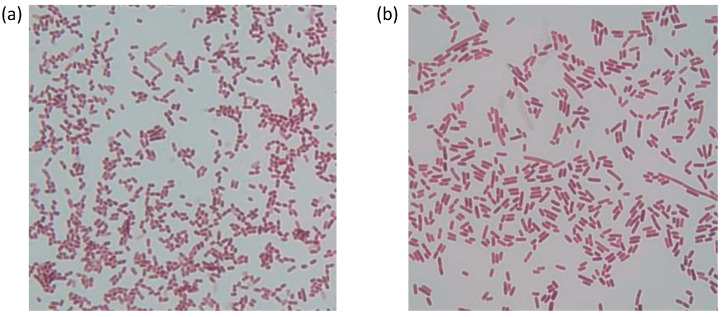
Gram staining of A4 (**a**) and A15 (**b**) bacteria showing both as gram variables. The images were photographed with an Olympus BX-40 light microscope. A 100× oil immersion objective was used with a WH10X/22 eyepiece.

**Figure 2 microorganisms-12-00070-f002:**
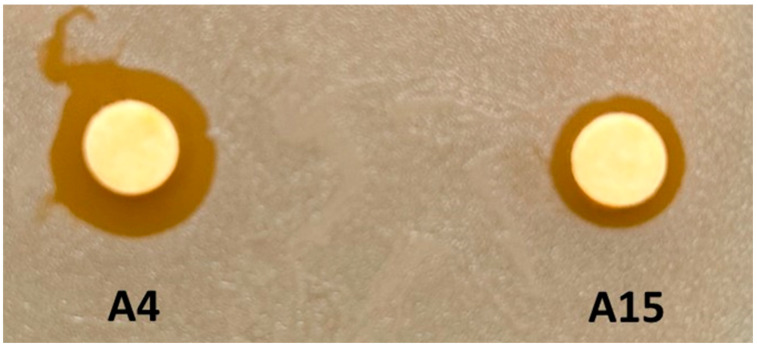
Growth inhibition spot assay with *Staphylococcus aureus* target on BHI media. The A4 and A15 isolates’ saturated discs are as designated in the figure.

**Figure 3 microorganisms-12-00070-f003:**
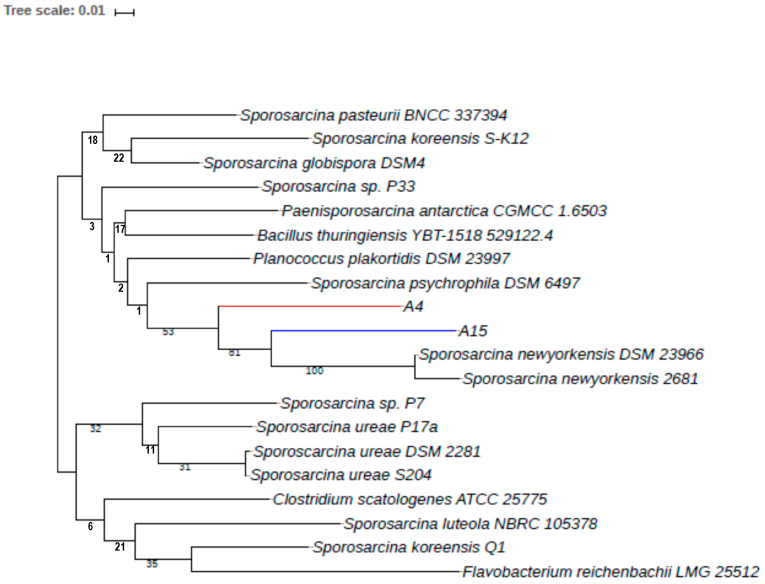
Minimum evolution tree inferred with FastME 2.1.6.1 [31] from Genome BLAST Distance Phylogeny (GBDP) distances calculated from genome sequences of strains A4, A15, and closest homologous genomes. The branch lengths are scaled in terms of GBDP distance formula d5 [47]. The numbers above branches are GBDP pseudo-bootstrap support values from 100 replications.

**Figure 4 microorganisms-12-00070-f004:**
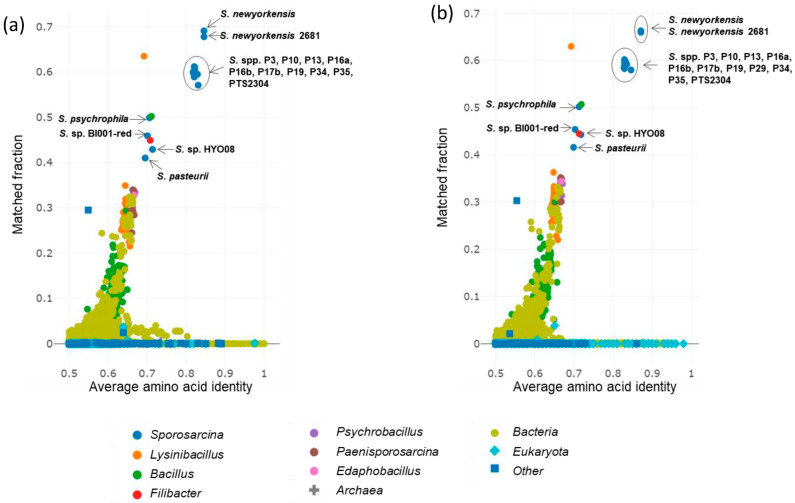
Average amino acid identity (AAI) of Uniprot species with strain A4 (**a**) and A15 (**b**). Species are grouped and colored by genus. Fraction of AAI vs. aligned matches of different reference species with query strains (A4 and A15) are plotted, showing *S. newyorkensis* as the closest species with A4 and A15 strains based on AAI (<88%). Genomes with AAI less than 0.5.

**Figure 5 microorganisms-12-00070-f005:**
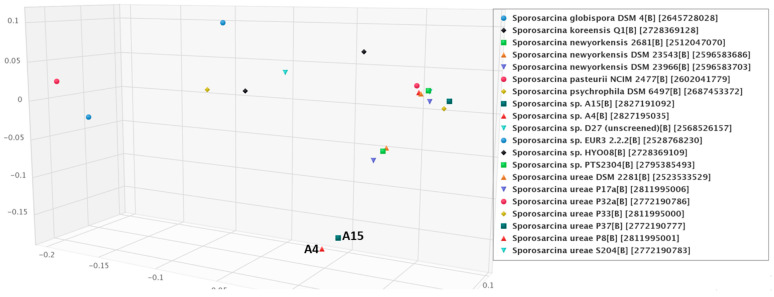
Principal coordinate analysis (PCoA) plot of COG profiles of *Sporosarcina* genomes based on Bray–Curtis distance matrix. Compositional dissimilarities among genomes were measured using Bray–Curtis dissimilarity coefficient of COG abundances.

**Table 1 microorganisms-12-00070-t001:** Average nucleotide identity (ANI) comparisons between A4, A15, and reference *Sporosarcina* spp. genomes.

Genome1	Genome2	ANI1->2	ANI2->1	AF1->2	AF2->1
A4	*Sporosarcina newyorkensis* DSM 23543	78.71	78.71	66.96	68.03
A4	*Sporosarcina newyorkensis* 2681	78.67	78.67	67.05	67.49
A4	*Sporosarcina newyorkensis* DSM 23966	78.65	78.65	70.14	71.90
A4	*Sporosarcina ureae* P37	77.38	77.38	58.77	65.46
A4	*Sporosarcina ureae* P33	77.35	77.35	58.64	67.62
A4	*Sporosarcina ureae* S204	76.52	76.52	57.93	64.57
A4	*Sporosarcina ureae* DSM 2281	76.49	76.49	58.85	65.85
A4	*Sporosarcina ureae* P17a	76.36	76.37	59.16	64.84
A4	*Sporosarcina ureae* P8	76.32	76.32	58.18	65.07
A4	*Sporosarcina ureae* P32a	76.23	76.23	59.58	65.83
A4	*Sporosarcina* sp. PTS2304	76.14	76.13	57.39	62.48
A4	*Sporosarcina pasteurii* ATCC 11859	72.32	72.32	0.05	73.76
A4	*Sporosarcina koreensis* Q1	71.27	71.26	39.81	34.46
A4	*Sporosarcina psychrophila* DSM 6497	70.80	70.79	40.98	34.54
A4	*Sporosarcina* sp. HYO08	70.79	70.77	36.45	43.88
A4	*Sporosarcina* sp. ZBG7A	70.65	70.65	36.44	38.27
A4	*Sporosarcina koreensis* S-K12	70.56	70.55	34.08	42.44
A4	*Sporosarcina* sp. D27 (unscreened)	70.53	70.53	36.94	38.24
A4	*Sporosarcina* sp. EUR3 2.2.2	69.03	69.04	24.35	26.61
A4	*Planococcus plakortidis* DSM 23997	68.84	68.84	23.73	26.88
A4	*Sporosarcina globispora* DSM 4	68.20	68.20	20.48	13.96
A4	*Sporosarcina pasteurii* NCIM 2477	67.62	67.62	15.48	13.72
A4	*Bacillus thuringiensis* YBT-1518	67.20	67.21	16.29	9.72
A4	*Clostridium scatologenes* ATCC 25775	65.38	65.38	4.98	2.54
A4	*Flavobacterium reichenbachii* LMG 25512	63.94	63.94	0.48	0.31
A4	A15	78.08	78.08	68.39	69.72
A15	*Sporosarcina newyorkensis* 2681	81.08	81.08	67.89	67.04
A15	*Sporosarcina newyorkensis* DSM 23966	81.06	81.06	69.65	69.99
A15	*Sporosarcina newyorkensis* DSM 23543	81.06	81.06	68.18	67.92
A15	*Sporosarcina* sp. PTS2304	77.27	77.27	60.90	64.83
A15	*Sporosarcina ureae* S204	77.09	77.10	59.43	64.98
A15	*Sporosarcina ureae* P8	77.08	77.08	60.34	66.20
A15	*Sporosarcina ureae* DSM 2281	77.07	77.07	60.28	66.16
A15	*Sporosarcina ureae* P17a	76.96	76.96	59.87	64.34
A15	*Sporosarcina ureae* P32a	76.95	76.95	60.72	65.85
A15	*Sporosarcina ureae* P33	76.89	76.91	59.64	67.56
A15	*Sporosarcina ureae* P37	76.88	76.88	60.14	65.70
A15	*Sporosarcina pasteurii* ATCC 11859	72.88	72.88	0.05	74.11
A15	*Sporosarcina* sp. HYO08	71.23	71.23	38.19	45.08
A15	*Sporosarcina psychrophila* DSM 6497	71.07	71.07	42.06	34.78
A15	*Sporosarcina koreensis* Q1	71.06	71.05	40.48	34.37
A15	*Sporosarcina* sp. ZBG7A	70.75	70.75	37.78	38.91
A15	*Sporosarcina* sp. D27 (unscreened)	70.68	70.68	37.84	38.42
A15	*Sporosarcina* sp. EUR3 2.2.2	69.64	69.64	26.63	28.59
A15	*Sporosarcina koreensis* S-K12	69.02	69.02	32.93	40.24
A15	*Bacillus thuringiensis* YBT-1518	67.88	67.89	17.47	10.25
A15	*Planococcus plakortidis* DSM 23997	67.87	67.87	23.39	25.99
A15	*Sporosarcina globispora* DSM 4	67.65	67.65	20.85	13.93
A15	*Sporosarcina pasteurii* NCIM 2477	67.22	67.22	15.66	13.61
A15	*Clostridium scatologenes* ATCC 25775	65.99	65.99	5.91	2.96
A15	*Flavobacterium reichenbachii* LMG 25512	64.45	64.45	0.47	0.30

**Table 2 microorganisms-12-00070-t002:** Pairwise comparisons of A4 and A15 genomes vs. closest *Sporosarcina* spp. genomes for Digital DNA/DNA hybridization and difference in G+C content.

Query Strain	Subject Strain	dDDH (d4, in %)	G+C Content Difference (in %)
A4	*Sporosarcina newyorkensis* 2681a	23.5	5.85
A4	*Sporosarcina* sp. P33	20.5	1.64
A4	*Sporosarcina ureae* P17a	20	4.67
A4	*Sporosarcina ureae* S204	19.4	4.66
A4	*Sporosarcina ureae* DSM 2281a	19.2	5.6
A4	*Sporosarcina* sp. P7	19.2	3.68
A4	*Sporosarcina newyorkensis* DSM 23966	15	3.99
A4	*Sporosarcina luteola* NBRC 105378	13.5	2.77
A4	*Sporosarcina koreensis* Q1a	13.4	2.95
A4	*Sporosarcina koreensis* S-K12	13.4	7.81
A4	*Sporosarcina psychrophila* DSM 6497	13.3	5.84
A4	*Sporosarcina pasteurii* BNCC 337394	13.3	7.01
A4	*Bacillus thuringiensis* YBT-1518 529122.4	13.1	10.76
A4	*Planococcus plakortidis* DSM 23997	13.1	3.8
A4	*Paenisporosarcina aarctica* CGMCC 1.6503	13.1	9.18
A4	*Sporosarcina globispora* DSM4	13.1	6.62
A4	*Clostridium scatologenes* ATCC 25775	13	16.57
A4	*Flavobacterium reichenbachii* LMG 25512a	12.9	8.8
A4	A15	24.2	0.36
A15	*Sporosarcina newyorkensis* 2681a	27.4	5.49
A15	*Sporosarcina* sp. P33	20.7	1.28
A15	*Sporosarcina ureae* S204	20.1	4.3
A15	*Sporoscarcina ureae* DSM 2281a	19.9	5.25
A15	*Sporosarcina* sp. P7	19.8	3.32
A15	*Sporosarcina ureae* P17a	19.5	4.31
A15	*Sporosarcina newyorkensis* DSM 23966	16.2	3.64
A15	*Sporosarcina luteola* NBRC 105378	13.6	2.41
A15	*Sporosarcina psychrophila* DSM 6497	13.5	5.48
A15	*Sporosarcina koreensis* Q1a	13.5	3.31
A15	*Sporosarcina pasteurii* BNCC 337394	13.3	6.65
A15	*Sporosarcina koreensis* S-K12	13.3	8.17
A15	*Bacillus thuringiensis* YBT-1518 529122.4	13.2	10.4
A15	*Paenisporosarcina aarctica* CGMCC 1.6503	13.2	8.82
A15	*Planococcus plakortidis* DSM 23997	13.2	4.16
A15	*Sporosarcina globispora* DSM4	13.1	6.26
A15	*Clostridium scatologenes* ATCC 25775	13	16.21
A15	*Flavobacterium reichenbachii* LMG 25512a	12.9	8.45

## Data Availability

The Whole Genome Shotgun project and 16S rRNA gene sequences have been deposited at DDBJ/ENA/GenBank. The accession numbers of strain A4 are WHVV01000000 (genome) and MN582949 (16S rRNA gene) and strain A15 are WHVW00000000 (genome) and MN582950 (16S rRNA gene).

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
