# Peer review of "Phenotypic Characterization and Draft Genome Sequence Analyses of Two Novel Endospore-Forming Sporosarcina spp. Isolated from Canada Goose (Branta canadensis) Feces"

_microorganisms, 2023, doi:10.3390/microorganisms12010070_

Round 1

Reviewer 1 Report

Comments and Suggestions for Authors

Dear Authors,

the manuscript presented to me for review is interesting and will certainly be interesting to readers. However, a few changes and corrections need to be made to make the manuscript more readable. My comments are included in the attachment.

Author Response

The review response is as:

L27: Please change from „Microbiome; Spore-forming bacteria; Urease; Probiotic” to “microbiome; spore-forming bacteria; urease; probiotic”.

The text as requested is now lower case

L61: Please precisely specify the purpose of the study

The text has been edited to: “Based on previous results [20], we hypothesized that the isolation of chloroform-resistant bacterial strains from Canada goose (Branta canadensis) feces would select for spore-forming bacteria of avian species.”

L62: There is no information about the consent of the Ethics Committee for the research - please complete it.

This is stated in the Institutional Review Board Statement: The Oregon State University Institutional Biosafety Committee has approved the research reported herein as IBC Proposal #3923. Lines 428-429

Although this statement is sadly missing for several published studies in the journal

L64: Please change from “44.0582° N, 121.3153° W” to “44.0582°N, 121.3153°W”.

The text has been edited as requested

L65: Please change to “-80°C’. The text has been edited as requested

L67: Please change from “15- or 50-ml” to “15 or 50 ml”.  The text has been edited as requested

L69: Please change to “1000”.L L69: The text has been edited as requested

L88: Change from “mL” to “ml”. The text has been edited as requested

L103: There is no explanation of what the abbreviation “OD” means. Please complete this.

OD was defined as Optical Density (OD)

L113: I suggest referring to a more recent publication (e.g., https://doi.org/10.1093/nar/gks808) rather than the one from 2002.

Yes, Klindworth et al., 2013 is a more recent publication on analyses of 16S rRNA primers. However, the PCR primers  used in our investigation are from the reference cited in the manuscript providing a near full-length 16S rRNA gene. Consequently, we want to maintain the congruence with our methods used to obtain the data.

L147: No appropriate description and legend for Figure S1. What do [A] and [B] mean? No markings for antibiotic discs - please complete. It's a pity that the photos differ in terms of quality (e.g., contrast, lighting) and frame. Please read the form of presentation of the figures included in the magazine's templet.

The Figure S1 legend has been edited with A and B panels denoted in the figure legend. The antibiotic discs are listed in the figure legend clockwise as reported in the figure legend.

L148-149: The photos are of poor quality. It's actually hard to notice anything. Please add better quality photos. Select the enlargement area and place the inset next to the original image. Add a note to the figure legend explaining what the inset shows and the resulting enlargement.

The images were taken with an Olympus BX-40 light microscope, so no true scale bar as with EM images. A 100X oil immersion objective along with a WH10X/22 eyepiece. The microscope and magnification text have been added to the Figure legend. Lines 149-151.

L161: No appropriate description and legend for Figure S2. Please add the appropriate markings on the image and explain in the legend which plate was with the appropriate medium, A4 and A15. Not every reader needs to have microbiological knowledge. This will certainly make it easier for all readers to interpret the results.

The figure legend and methods have been edited to explain which plate was with the appropriate medium, A4 and A15.

L167: Please put a dot between the sentence “..digest starch [52] Most metabolic..”.

A period was added to end the sentence.

L202: What was the magnification of image, scale bar?L

There was no magnification for the figure and no scale bar was utilized to photograph the image. The method was as described in reference 32 and our previous publication: https://www.mdpi.com/2673-8007/3/4/77

L277-281: Please change the images to sharper ones. Please present the figures and description according to the templet. No explanation for (b) - please complete. S. newyorkensis should be in italics. Move Figure after paragraph L282-296.

I have attempted to move the figure as suggested by the reviewer. However, the template provided by the journal has not allowed me to do so after repeated attempts. This can be completed by the editorial staff if accepted for publication and required by the editorial staff. S. newyorkensis is now in italics

L339: Please change the image to a more sharp

NOTE: I have provided the journal with a pdf of the figures that have a sharper image. I have maintained the original figures in the manuscript to avoid disruption of the text in the current revised version.

L361-362: Sporosarcina should be in italics. Corrected for all genus listings in the text.

L372: This is a summary of the results, not a conclusion. Please move to the previous section as a summary of the results.

Summary statement has been removed and stated as pert of the discussion.

-I know this section (Conclusion) is not mandatory, but it should be added to the manuscript because the discussion is unusually long and complex. Please write conclusions that support the hypothesis and aim of the study.

The conclusion section was included in the manuscript.

L445-597: Please correct the references according to the instructions (templet).

References have been prepared to meet the journal format.

Reviewer 2 Report

Comments and Suggestions for Authors

The subject is interesting and treated responsibly by the authors. However, some improvements are needed. 

The aim of this study should be mentioned in the introduction.

The tables should be on one page only (place table 2).

Conclusions should be reworded.

Author Response

The review response is as:

The aim of this study should be mentioned in the introduction.

Lines 59-61 state the hypothesis for the reported investigation.

The tables should be on one page only (place table 2).

Tables are now on one page of the manuscript.

Conclusions should be reworded.

We are uncertain as how to “reword” the conclusions. We reported in the conclusions that the results reported support the hypothesis that the A4 and A15 isolates are unique among the Sporosarcina spp. The conclusion section includes statements based on the results of the investigations involving the phenotypic and genomics characterizations of the two bacterial isolates.

Reviewer 3 Report

Comments and Suggestions for Authors

I read this manuscript. It fits the scope of this journal. After careful reading, this manuscript needs a major revision before the conclusion regarding acceptance or rejection. I encourage the authors to address the following comments.

Comment # 1: Line 13-26: The abstract lacks its main structures (objectives, clear methodologies, findings, and conclusion). Please address this issue.

Comment # 2: Line 57-62: I understand your rationale, but you should try to strengthen it. Please clarify your objectives, too.

Comment # 3: Line 62-137: in the methodology section, I recommend the authors to address these points:

1-               The number of samples/replicates were used?

2-               All relevant information regarding geese (age, ……)

3-               When were the trials conducted?

4-               Ethical code to use the animals?

5-               Provide the statistical approach to present your data (if it is available)

Comment # 4: For the result section, some figures and table legends lack important information such as the number of samples/animals/replicates, probability of significance, and reference to the main findings. Also, some tables were not provided in the manuscript (may be supplemented files). Please provide an access to it.

Comment # 5: Line 372-406: The conclusion section is very long. Please try to be more specific.

Author Response

Comment # 1: Line 13-26: The abstract lacks its main structures (objectives, clear methodologies, findings, and conclusion). Please address this issue.

We kindly disagree with this statement. The text is written to state that “Two Gram-variable, spore forming, rod-shaped aerobic bacteria designated as strain A4 and A15, were isolated from feces of Canada geese…” and states initial phenotypic results. The abstract text reports 16S rRNA and summarized data for the analyses of the draft genomes for both bacterial isolates. Finally, it is concluded that “these bacteria are two novel isolates of the Sporosarcina genus” in the final sentence of the abstract.

Comment # 3: Line 62-137: in the methodology section, I recommend the authors to address these points:

1-The number of samples/replicates were used?

We isolated two separate bacteria from “Fresh feces from geese…” as described in the methods.

2-All relevant information regarding geese (age, ……)

We isolated materials obtained from “free-ranging geese…” that as been added to the text L63. The age of these animals is unknown and are essentially environmental samples obtained after geese defecated on site, L64.

3-When were the trials conducted?

That information was added in L65.

4-Ethical code to use the animals?

No animals were used during this and ongoing investigations and therefore exempt from our organizations Institutional Animal Care And Use Committee. We do state in the Institutional Review Board Statement: The Oregon State University Institutional Biosafety Committee has approved the research reported herein as IBC Proposal #3923 in Lines 428-429.

5-Provide the statistical approach to present your data (if it is available)

All statistical analyses are stated in the Methods section such as the “Tukey honest significant difference (HSD)…” for the phenotype array data in L102. Genomics statistics for comparison of our reported genomes are reported in L126-137.

Comment # 4: For the result section, some figures and table legends lack important information such as the number of samples/animals/replicates, probability of significance, and reference to the main findings. Also, some tables were not provided in the manuscript (may be supplemented files). Please provide an access to it.

All data as analyzed is reported in the methods section. Supplemental figures were and are provided as supplemental information. Figures and tables in that zip file have been updated as well.

Comment # 5: Line 372-406: The conclusion section is very long. Please try to be more specific.

We have provided stated conclusions regarding the hypothesis that the reported bacterial isolates are unique members of the genus Sporosarcina. This required a more extensive conclusions section.

Round 2

Reviewer 3 Report

Comments and Suggestions for Authors

Thank you very much for your replies. Indeed, I found the authors did not respond to ALL comments raised in the first round of reviewing.

1-Please reorganize the abstract to represent the content of the study (clear objectives and methodologies,.........).

2-As the result section, materials & method section SHOULD be organized into steps (headings and subheadings). Please, state the statistical section, separately. A figure representing the research design could be helpful. 

3-I don't agree with your reply about the conclusion section. You could  state such information under subtitle "Summary & conclusion" or summarize the conclusion section.

Author Response

microorganisms-2703941 review 2

Thank you very much for your replies. Indeed, I found the authors did not respond to ALL comments raised in the first round of reviewing.

1-Please reorganize the abstract to represent the content of the study (clear objectives and methodologies,.........).

Text was added to the first sentence of the abstract to address this edit request. If more text is added it will exceed the 200-word limit for abstracts as required by the journal. Also, a review of similar publications in the journal demonstrates that the abstract as written is in the required style. Please refer to Alotaibi F et al.:

https://www.mdpi.com/2076-2607/11/12/2972

2-As the result section, materials & method section SHOULD be organized into steps (headings and subheadings). Please, state the statistical section, separately. A figure representing the research design could be helpful.

The Methods section as the Results section is divided into subheadings. We have kept the statistical methods to be congruent with each analyses, e.g., the statistics used for the phenotype microarrays and the genomic principal coordinates and phylogenetic analyses are reported with each method.

3-I don't agree with your reply about the conclusion section. You could state such information under subtitle "Summary & conclusion" or summarize the conclusion section.

The subtitle has been edited to “Summary and Conclusion”
